# Central nicotine induces browning through hypothalamic κ opioid receptor

Patricia Seoane-Collazo [1,2], Laura Liñares-Pose [1,2,12], Eva Rial-Pensado[1,2,12], Amparo Romero-Picó[1,2,12], José María Moreno-Navarrete[2,3], Noelia Martínez-Sánchez [1,4], Pablo Garrido-Gil [5,6], Ramón Iglesias-Rey[7], Donald A. Morgan[8], Naoki Tomasini[9], Samuel Andrew Malone [4], Ana Senra[1,2], Cintia Folgueira [1,2], Gema Medina-Gomez[10], Tomás Sobrino [7], José L. Labandeira-García[5,6], Rubén Nogueiras [1,2], Ana I. Domingos[4], José-Manuel Fernández-Real[2,3], Kamal Rahmouni[8,11], Carlos Diéguez[1,2] & Miguel López [1,2]

Increased body weight is a major factor that interferes with smoking cessation. Nicotine, the main bioactive compound in tobacco, has been demonstrated to have an impact on energy balance, since it affects both feeding and energy expenditure at the central level. Among the central actions of nicotine on body weight, much attention has been focused on its effect on brown adipose tissue (BAT) thermogenesis, though its effect on browning of white adipose tissue (WAT) is unclear. Here, we show that nicotine induces the browning of WAT through a central mechanism and that this effect is dependent on the κ opioid receptor (KOR), specifically in the lateral hypothalamic area (LHA). Consistent with these findings, smokers show higher levels of uncoupling protein 1 (UCP1) expression in WAT, which correlates with smoking status. These data demonstrate that central nicotine-induced modulation of WAT browning may be a target against human obesity.

---

[1] Department of Physiology, CiMUS, University of Santiago de Compostela-Instituto de Investigación Sanitaria, 15782 Santiago de Compostela, Spain. [2] CIBER Fisiopatología de la Obesidad y Nutrición (CIBEROBN), 28029 Santiago de Compostela, Spain. [3] Department of Diabetes, Endocrinology and Nutrition, Hospital de Girona Dr Josep Trueta, Institut D'investigació Biomèdica de Girona (IdIBGi) and University of Girona, 17007 Girona, Spain. [4] Department of Physiology, Anatomy and Genetics, University of Oxford, Oxford OX1 3PT, UK. [5] Department of Morphological Sciences, CiMUS, University of Santiago de Compostela-Instituto de Investigación Sanitaria, 15782 Santiago de Compostela, Spain. [6] CIBER Enfermedades Neurodegenerativas (CIBERNED), 28029 Santiago de Compostela, Spain. [7] Clinical Neurosciences Research Laboratory, Instituto de Investigación Sanitaria, 15782 Santiago de Compostela, Spain. [8] Department of Pharmacology, University of Iowa, Iowa City, IA 52242, USA. [9] Obesity Laboratory, Instituto Gulbenkian de Ciência, 2780-156 Oeiras, Portugal. [10] Department of Basic Sciences of Health, Area of Biochemistry and Molecular Biology, Universidad Rey Juan Carlos, 28922 Alcorcon-Madrid, Spain. [11] Department of Internal Medicine, University of Iowa, Iowa City, IA 52242, USA. [12] These authors contributed equally: Laura Liñares-Pose, Eva Rial-Pensado, Amparo Romero-Picó. Correspondence and requests for materials should be addressed to C.Dég. (email: carlos.dieguez@usc.es) or to M.Lóp. (email: m.lopez@usc.es)

Nicotine, the main bioactive molecule in tobacco, is an exogenous ligand of the nicotinic acetylcholine receptor (nAChR), and it influences various functions in the central nervous system (CNS) including energy homeostasis[1–5]. Consequently, nicotine has been demonstrated to decrease food intake through the modulation of hypothalamic neuropeptide systems[1,3,5,6]. In addition, nicotine promotes energy expenditure (EE) by activating BAT thermogenesis through a mechanism that involves hypothalamic inhibition of AMP-activated protein kinase (AMPK) and an increase in the sympathetic nervous system (SNS) tone[3,7,8]. Current evidence has also pointed to the expression of the cholinergic receptor nicotinic alpha 2 subunit (Chrna2) in white adipose tissue (WAT) during the activation of beige fat and that α2-nAChR increases thermogenesis in UCP1-positive beige adipocytes[9]. However, despite these findings, whether nicotine may act centrally to regulate browning of WAT and, more importantly, the mechanisms underlying this effect, remain unclear.

Nicotine can modulate the hedonic/reward pathways, such as the endogenous opioid system[10], which includes the opioid peptides endorphins, enkephalins, dynorphins and endomorphins that act as ligands of the μ, δ, and κ opioid receptors (MOR, DOR, and KOR), a family of G-couple protein receptors that are widely distributed throughout the CNS[11–13]. The opioid system has been established as an important regulator in neural hedonic and reward processes, such as those leading to addictive behaviors[11,13–15]. This is an important issue given the existence of cross addiction, people swapping from one addiction to another, e.g., nicotine to food addiction[16]. Notably, current data also point to the opioid system as a homeostatic regulator of energy balance. Naltrexone, an opioid receptor antagonist, is now approved in combination with bupropion for the treatment of obesity[17–19]. Moreover, the opioid system has been proposed as a possible target for smoke cessation[20,21]. At the central level, the opioid system can act in multiple brain areas. For example, it is known that dynorphin (DYN, an endogenous ligand of KOR) modulates food intake and fat mass by increasing SNS activity[22] and that hypothalamic KOR signaling mediates the orexigenic action of ghrelin[23] and melanin-concentrating hormone (MCH)[24]. In addition, mice lacking MOR, KOR, or DOR resist the development of diet-induced obesity, despite hyperphagia, due to the increased energy expenditure, leading to an overall improvement of the metabolic phenotype[25–27]. Based on this evidence, we hypothesized that nicotine's action on energy homeostasis could be mediated, at least in part, by the opioid signaling. We found that KOR, specifically in the lateral hypothalamus (LHA), is necessary for nicotine's action on body weight, BAT thermogenesis and notably browning of WAT. The clinical relevance of our data is demonstrated by the fact that smokers displayed higher expression of UCP1 in the WAT and a correlation with the smoking status. Our data suggest a previously unknown link between KOR in the LHA in mediating nicotine-induced energy expenditure.

## Results

**Peripheral nicotine induces browning in WAT.** First, we aimed to investigate the effect of peripheral (subcutaneous, SC) nicotine treatment on energy balance in rats. Nicotine induced marked body weight reduction and anorectic effects (Supplementary Fig. 1a, b), associated with higher EE, elevated locomotor activity (LA) and reduced respiratory quotient (RQ) (Supplementary Fig. 1c–e), as well as increased UCP1 expression in BAT (Supplementary Fig. 1f). Next, we examined the ability of nicotine to induce the browning process in different WAT depots. We found that SC administration of the drug for one week promoted UCP1 expression and/or

reduced adipocyte area in the gonadal (gWAT), subcutaneous (sWAT), and visceral (vWAT) WAT (Supplementary Fig. 1g–l), all of which indicate the occurrence of browning.

**Central nicotine induces browning in WAT.** Next, we examined whether central nicotine might recapitulate the action of peripheral administration on WAT browning. In mice, intra-cerebroventricular (ICV) injection of nicotine for 7 days caused a marked anorexia and weight loss (Fig. 1a, b), as well as reduced adiposity (Fig. 1c). Analysis of WAT demonstrated that central nicotine induced a robust increase in the expression of thermogenic markers in gWAT (Fig. 1d), which was indicative of browning. To further characterize that effect, we histologically analyzed the different WAT depots after central nicotine administration. Our data showed increased UCP1 immunoreactivity, associated with reduced adipocyte area, in all fat depots analyzed (Fig. 1e–j), confirming the existence of browning.

The importance of the SNS in the control of UCP1 expression and in eliciting browning led us to consider the possibility that nicotine may induce sympathetic activation of the nerves subserving WAT. Consistent with such possibility, we found that ICV administration of nicotine markedly increased sWAT sympathetic nerve traffic recorded directly by microneurography (Fig. 2a, b). Notably, the increase in WAT sympathetic nerve traffic evoked by ICV nicotine was dose-related, with sympathetic activity increasing by 66.7 ± 26.5% ($P < 0.01$) and 227.6 ± 51.5% ($P < 0.001$) 4 h after 0.15 μg and 0.3 μg ICV nicotine, respectively. In keeping with this, central nicotine induced a significant increase in the norepinephrine (NE) concentration in sWAT and vWAT (Fig. 2c). To further characterize the ability of nicotine to modulate the sympathetic tone on WAT, we investigated its impact on the density of the SNS fibers innervating gWAT pads. By using tyroxine hydroxylase (TH) immunostaining coupled to *Adipo-Clear*[28,29], our data showed that central treatment with nicotine promoted a clear trend to increase the length, area and volume of the dendrites innervating gWAT (Fig. 2d and Supplementary Movie 1), which was indicative that besides promoting functional changes in sympathetic activity, nicotine also elicited changes in the plasticity and architecture of sympathetic fibers. Overall, these data indicate that central administered nicotine exerted its action on WAT through activation of the SNS.

**KOR is required for nicotine action on energy balance.** The central opioid system is known to modulate energy balance. For instance, KOR blockade protected mice against the metabolic outcomes of obesity by increasing BAT thermogenesis and consequently EE[26]. However, whether KOR may impact and/or mediate the actions of nicotine on browning of WAT remains unexplored. Thus, we used a more robust experimental approach by examining the effect of nicotine on KOR null mice and their wildtype (WT) controls. In WT, but not in KOR null mice, nicotine decreased body weight (Fig. 3a, b) and adiposity (Fig. 3c–f). These changes were associated with increased EE (Fig. 3g, h) in WT but not in KOR null animals. Overall, this evidence indicates that KOR was needed for nicotine to exert its actions on energy balance. Notably, those effects were not explained by abnormalities in nicotinic receptors in the KOR null mice, because none of nicotinic receptors analyzed (β2-nAChR, β4-nAChR, α3-nAChR, α4-nAChR, α7-nAChR) showed major changes (only a slight increase in β2-nAChR) in their protein levels when compared to mice (Supplementary Fig. 2a). Moreover, we analyzed the concentration of dopamine (DA), 3,4-dihydroxyphenylacetic acid (DOPAC), serotonin (5-HT) and NE

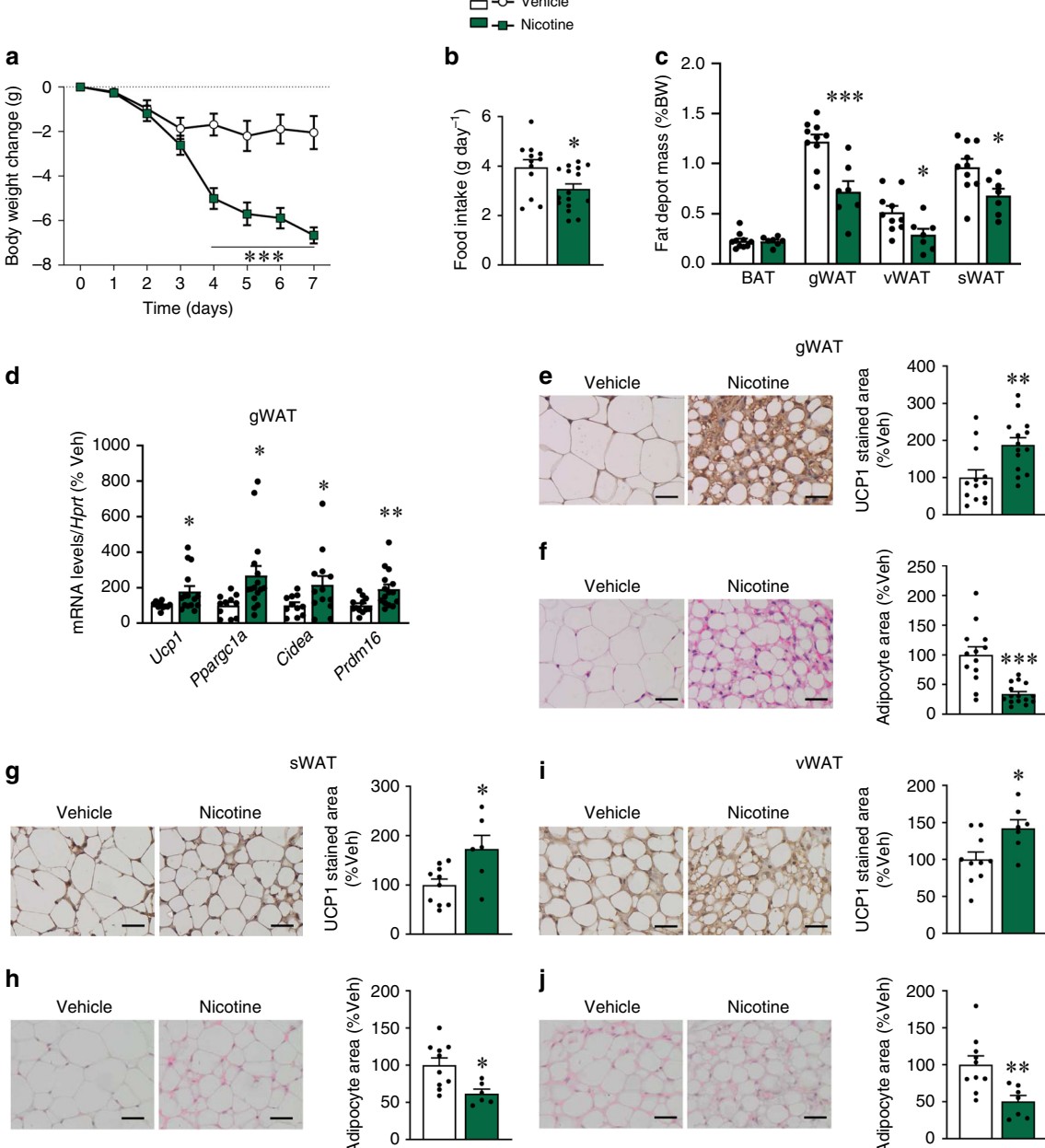

**Fig. 1** Effect of central nicotine WAT browning. **a** Body weight change (vehicle $n = 12$, nicotine $n = 16$ mice). **b** Daily food intake (vehicle $n = 12$, nicotine $n = 16$ mice). **c** Fat depot mass (vehicle $n = 10$, nicotine $n = 7$ mice). **d** mRNA levels of thermogenic markers in the gWAT (vehicle $n = 11$, 11, 11, 12, nicotine $n = 15$, 16, 13, 15 mice). **e** UCP1 staining (left panel 40×, scale bar: 100 µm) and UCP1 stained area (right panel) in gWAT (vehicle $n = 13$, nicotine $n = 14$ mice). **f** H&E staining (left panel 40×, scale bar: 100 µm) and adipocyte area (right panel) in gWAT (vehicle $n = 13$, nicotine $n = 15$ mice). **g** UCP1 staining (left panel 40×, scale bar: 100 µm) and UCP1 stained area (right panel) in sWAT (vehicle $n = 10$, nicotine $n = 6$ mice). **h** H&E staining (left panel 40×, scale bar: 100 µm) and adipocyte area (right panel) in sWAT (vehicle $n = 10$, nicotine $n = 6$ mice). **i** UCP1 staining (left panel 40×, scale bar: 100 µm) and UCP1 stained area (right panel) in vWAT (vehicle $n = 10$, nicotine $n = 7$ mice). **j** H&E staining (left panel 40×, scale bar: 100 µm) and adipocyte area (right panel) in vWAT (vehicle $n = 10$, nicotine $n = 7$ mice) of mice ICV treated with vehicle or nicotine. Center values represent average; error bars represent SEM. Statistical significance was determined by two-sided $t$-Student; *$P < 0.05$, **$P < 0.01$, ***$P < 0.001$ vs. vehicle. The experiments were repeated three times; the samples represent biological replicates. Source data are provided as a Source Data file

in the hypothalamus, striatum and cortex of WT and KOR KO mice. Our data show similar values for all the analyzed parameters (Supplementary Fig. 2b). Overall, this evidence suggested that the observed changes after nicotine treatment are not related to any generalized defect(s) that might characterize KOR mutant mice.

Further analysis demonstrated that nicotine induced the expression of UCP1 and other thermogenic markers in BAT of WT but not KOR null mice (Fig. 4a–d). Next, we

investigated whether nicotine actions on WAT browning might depend on KOR. Nicotine increased the UCP1 staining and reduced the adipocyte area in WT, but not in KOR null mice (Fig. 4e–h) in gWAT. In keeping with these data, the mRNA expression of browning markers, such as *Ucp1*, *Ppargc1a*, *Cidea*, and *Prdm16* was induced by nicotine in the gWAT of WT but not in KOR null animals (Fig. 4i, j). Altogether these results indicate that nicotine action on browning depends on KOR.

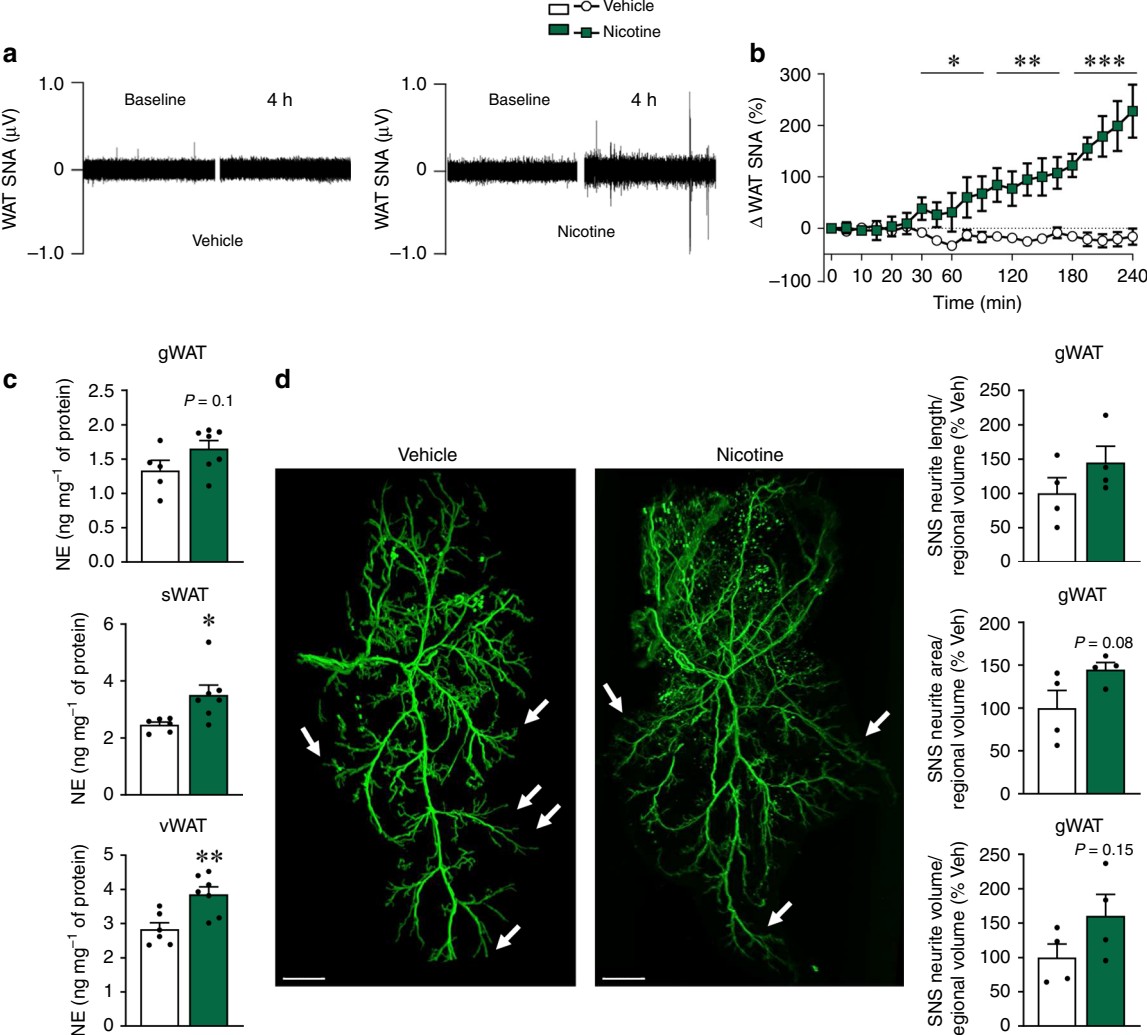

**Fig. 2** Effect of central nicotine on SNS. **a–b** Sympathetic nerve activity (SNA) recorded from the nerves subserving sWAT (vehicle $n = 6$, nicotine $n = 5$ mice). **c** NE levels in gWAT, sWAT, and vWAT (vehicle $n = 5, 6, 6$, nicotine $n = 7, 7, 7$ mice). **d** TH immunohistochemistry (right panel, scale bar: 1000 µm) and quantification of SNS neurite length, area and volume (left panel) in gWAT (vehicle $n = 4$, nicotine $n = 4$ mice). White arrows represent dendrites of mice treated ICV with vehicle or nicotine. Center values represent average; error bars represent SEM. Statistical significance was determined by two-sided $t$-Student; $*P < 0.05$, $**P < 0.01$, $***P < 0.001$ vs. vehicle. Source data are provided as a Source Data file

**KOR in the LHA mediates nicotine action on energy balance.** Next, we assessed the specific brain area where KOR mediates the effect of central nicotine. As previously reported[26], our mRNA analyses demonstrated negligible expression of KOR in the WAT and BAT of both rats and mice, when compared with hypothalamus (Supplementary Fig. 2c, d). It has been shown that hypothalamic KOR is involved in the modulation of feeding[22,23], but no evidence has linked hypothalamic KOR with the actions of nicotine. To address this, we selectively silenced the expression of KOR in hypothalamic sites that are known to modulate SNS activity and/or browning, namely the ventromedial nucleus (VMH) and the LHA[30–33] in rats using adeno-associated virus (AAV) harboring a shRNA against *Opkr1* (shRNA-*Opkr1*)[23]. The efficiency of the knockdown was validated by the decreased *Opkr1* mRNA expression and/or KOR (but not MOR or DOR) protein content in the VMH and LHA (Supplementary Fig. 2e–h). Knockdown of KOR in the VMH did not impact the effect of nicotine on body weight in rats (Supplementary Fig. 3a, b). Recent evidence demonstrated that nicotine actions on BAT are mediated through AMPK in the VMH[3]. However, our data showed that nicotine inhibitory effect on VMH AMPK is not

affected by selective knockdown of *Opkr1* in this nucleus (Supplementary Fig. 3c, d), suggesting that a KOR-AMPK interaction at this level is unlikely to be involved in the effect of central nicotine on fat depots. In keeping with these results, molecular analysis of BAT and WAT also demonstrated that the partial genetic ablation of KOR in the VMH did not impact the ability of nicotine to induce either BAT thermogenesis (Supplementary Fig. 3e–h) or the browning of WAT (Supplementary Fig. 3i–n).

In contrast to the VMH, when given into the LHA, AAV shRNA-*Opkr1* totally blunted the effect of nicotine on body weight (Fig. 5a, b), BAT thermogenesis (Fig. 5c–f) and the browning of gWAT, as demonstrated by normal UCP1 immunoreactivity (Fig. 5g, h), unaltered adipocyte area (Fig. 5i, j) and the expression of thermogenic markers (Fig. 5k, l), when compared to controls. Altogether, these results indicate that KOR in the LHA is required for the central nicotine actions on energy balance, BAT thermogenesis and browning of WAT.

**UCP1 in WAT is positively correlated with smoking in humans.** Finally, we analyzed the relationship between smoking

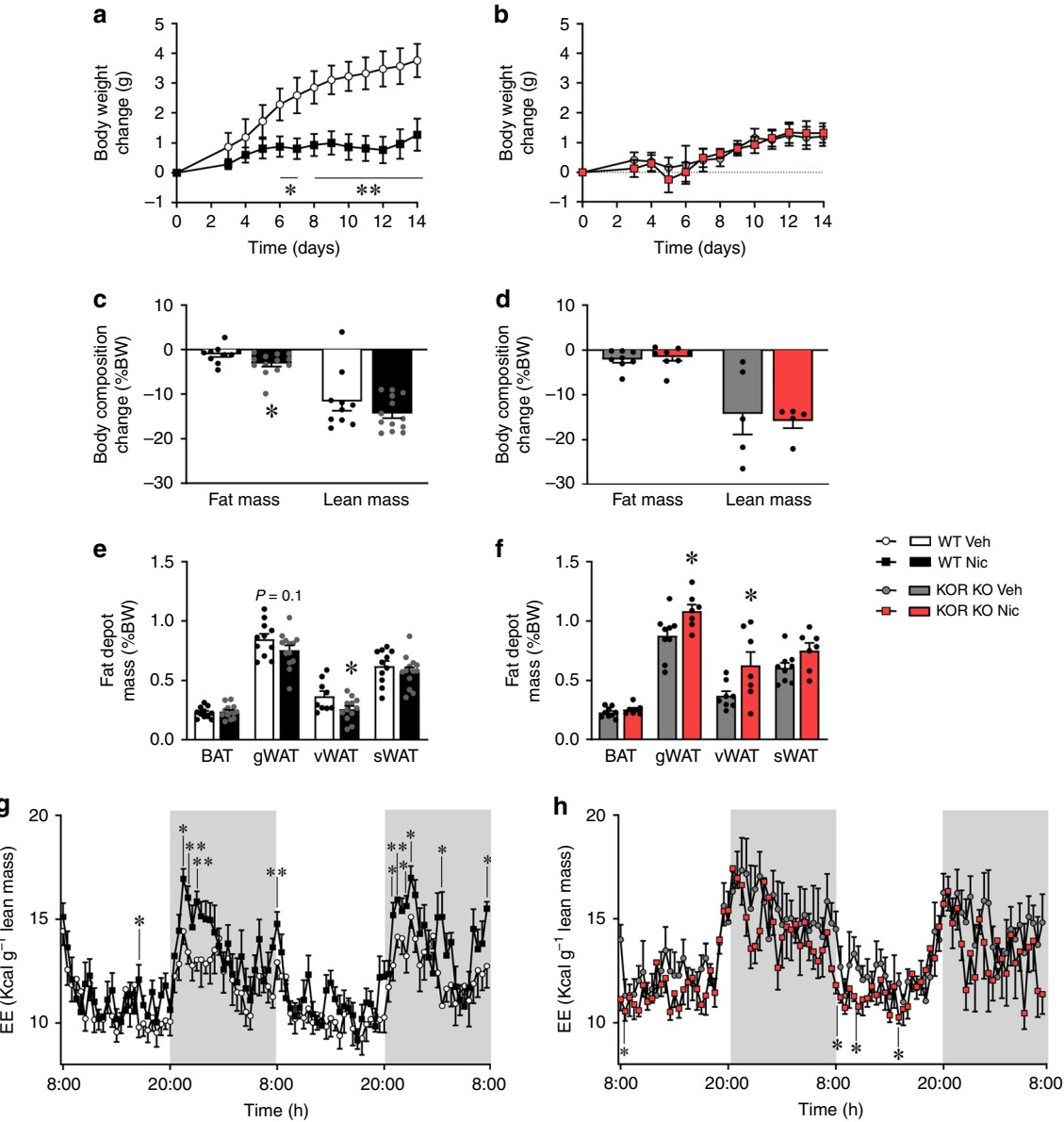

**Fig. 3** Effect of nicotine on energy balance in KOR null mice. **a–b** Body weight change (WT vehicle $n = 12$, WT nicotine $n = 14$, KO vehicle $n = 14$, KO nicotine $n = 13$ mice). **c–d** Body composition (Fat mass WT vehicle $n = 10$, WT nicotine $n = 13$, KO vehicle $n = 8$, KO nicotine $n = 8$; Lean mass WT vehicle $n = 10$, WT nicotine $n = 13$, KO vehicle $n = 5$, KO nicotine $n = 5$ mice). **e–f** Fat pads mass (WT vehicle $n = 11, 11, 9$ 11, WT nicotine $n = 13, 13, 12, 13$, KO vehicle $n = 9, 9, 8, 9$, KO nicotine $n = 7, 7, 7, 7$ mice). **g–h** Energy expenditure (EE) (WT vehicle $n = 5$, WT nicotine $n = 6$, KO vehicle $n = 5$, KO nicotine $n = 5$ mice) of WT (**a**, **c**, **e**, and **g**) or KOR KO (**b**, **d**, **f**, and **h**) mice treated with vehicle or nicotine. Center values represent average; error bars represent SEM. Statistical significance was determined by two-sided t-Student; *$P < 0.05$, **$P < 0.01$ vs. vehicle. The experiments were repeated two times; the samples represent biological replicates. Source data are provided as a Source Data file

and WAT UCP1 mRNA expression in humans (cohorts 1 and 2; Supplementary Table 1). In cohort 1, our data showed that the mRNA levels of UCP1 in sWAT were increased in smokers when compared to controls (Fig. 6a), despite both groups being similar in age and BMI (Fig. 6b, c; Supplementary Table 1). Notably, UCP1 expression correlated with smoking status, determined by the number of cigarettes per day (Fig. 6d). However, it is well known that smoking status is not always reliable when based merely on a questionnaire. In an independent series of 56 subjects (cohort 2), this association was confirmed by measuring the levels of circulating cotinine, an alkaloid found in tobacco that is the predominant metabolite of nicotine and is used as a biomarker for exposure to tobacco smoke[34,35]. Serum cotinine contributed significantly to 8% of the variance in age-adjusted and

BMI-adjusted UCP1 gene expression in vWAT (Fig. 6e and Supplementary Table 2). Overall, this evidence suggest that nicotine is also inducing browning in humans.

## Discussion

Data gleaned in recent years have shown that, in addition to BAT activation, browning of adipose tissue plays a protective role against diet-induced obesity[31,36,37]. Thus, increased energy expenditure via browning of WAT is now considered as a therapeutic target for obesity[31,36,37]. Therefore, there is a great deal of interest in uncovering the mechanisms governing this phenomenon. Available evidence indicate that browning is regulated by factors such as endocrine signals, for example thyroid hormones,

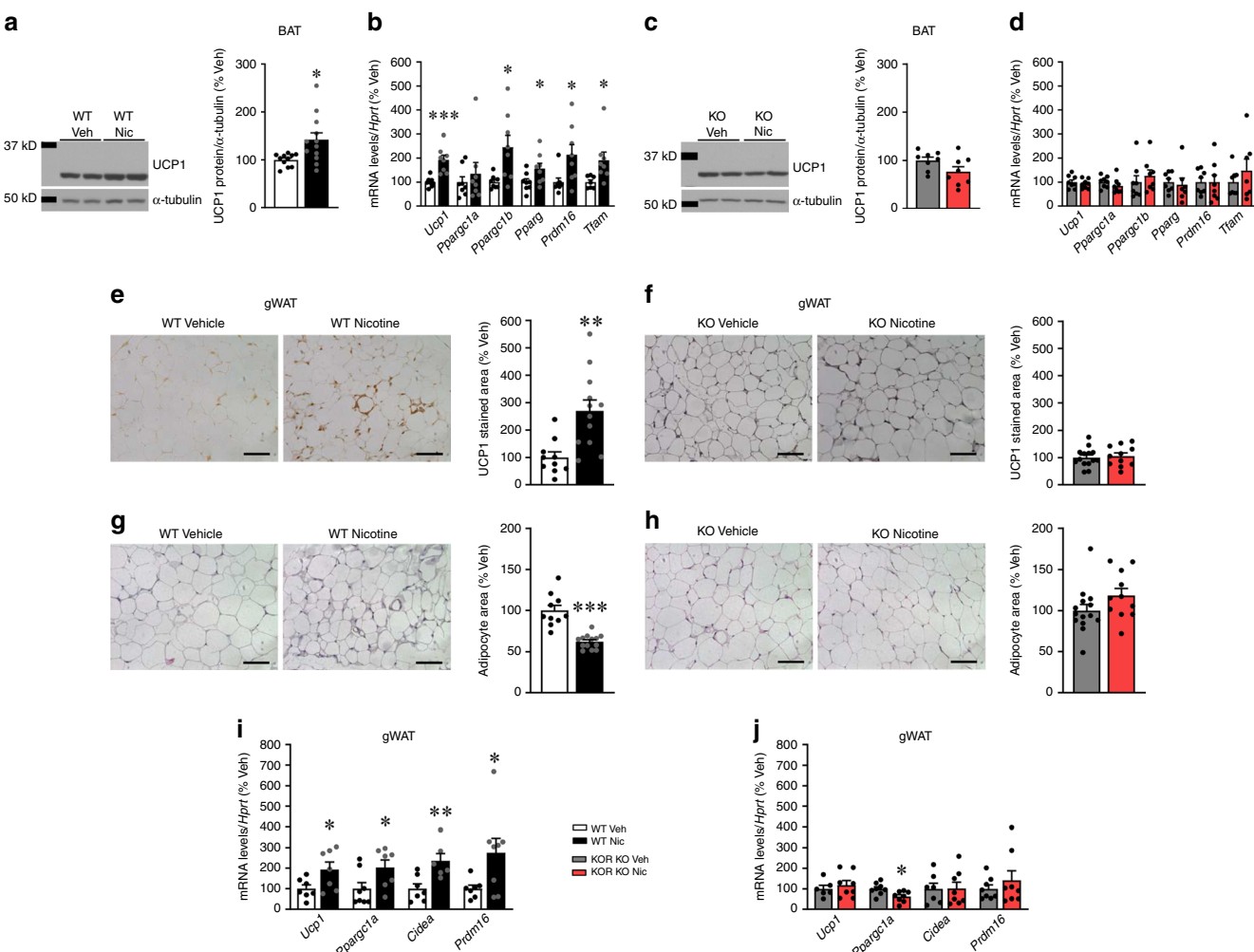

**Fig. 4** Effect of nicotine on BAT and WAT in KOR null mice. **a–c** Representative images (left panels) and protein levels of UCP1 in the BAT (right panels) (WT vehicle $n = 11$, WT nicotine $n = 12$, KO vehicle $n = 8$, KO nicotine $n = 8$ mice). **b–d** mRNA levels of thermogenic markers in the BAT (WT vehicle $n =$ 8, 8, 8, 8, 8, 7, WT nicotine $n = 8$, KO vehicle $n = 8$, KO nicotine $n = 8$, 8, 8, 7, 8, 7 mice). **e–f** UCP1 staining (left panels 40×, scale bar: 100 μm) and UCP1 stained area (right panels) in gWAT (WT vehicle $n = 10$, WT nicotine $n = 12$, KO vehicle $n = 14$, KO nicotine $n = 11$ mice). **g–h** H&E staining (left panels 40×, scale bar: 100 μm) and adipocyte area (right panels) in gWAT (WT vehicle $n = 10$, WT nicotine $n = 13$, KO vehicle $n = 14$, KO nicotine $n = 11$ mice). **i–j** mRNA levels of thermogenic markers in the gWAT (WT vehicle $n = 7$, 7, 7, WT nicotine $n = 7$, 7, 6, 8, KO vehicle $n = 6$, 8, 7, 8, KO nicotine $n = 8$, 8, 8, 8 mice) of WT (**a**, **b**, **e**, **g** and **i**) or KOR KO (**c**, **d**, **f**, **h** and **j**) mice treated with vehicle or nicotine. Center values represent average; error bars represent SEM. Statistical significance was determined by two-sided $t$-Student; *$P < 0.05$, **$P < 0.01$, ***$P < 0.001$ vs. vehicle. The experiments were repeated two times; the samples represent biological replicates. In the panels **a** and **c** showing images of gels, all the bands for each picture come from the same gel but they have been spliced for clarity. Source data are provided as a Source Data file

growth factors, local tissue microenvironment and sympathetic innervation[31,36–38]. Their influence is variable among the different WAT depots, but a common feature is that UCP1 expression and decrease volume specify the metabolic phenotype of this browning. A current challenging question is to uncover the mechanisms subserving browning and more specifically the ones related to sympathetic innervation since the degree of innervation of WAT directly correlates with browning and also with nor-epinephrine turnover[31,36,37].

Nicotine acts on energy balance through the modulation of food intake and energy expenditure[1–5,39]. Classical studies showed that daily injections of nicotine during 6 months led to induction of UCP1 in WAT of mice[40]. It should be noted, however, that overwhelming evidence indicate that (1) browning is a centrally induced process and it requires the SNS[33,36,41] and (2) the brain is the main target of nicotine actions in terms of energy metabolism, by modulating both feeding and BAT

thermogenesis[1–5]. Moreover, in humans, nicotine is mainly inhaled (smoking) and, in a lesser way, chewed. Following inhalation, nicotine rapidly reaches the brain—in less than eight seconds—and when chewed in less than 3–5 min[34,35]. Hence, the importance of using the central delivery to mimic/develop an experimental paradigm relevant for human disease. Therefore, we aimed to investigate whether nicotine might act in the CNS to control the browning of WAT. Our data showed that nicotine, when given ICV, reduced body weight and increased BAT ther-mogenesis and WAT browning to a similar extent as when given peripherally. Thus, nicotine markedly increased the UCP1 expression and/or immunoreactivity in all the examined WAT depots, as well as inducing morphological changes that were clearly indicative of browning. Considering this, it is likely that the negative energy balance induced by nicotine, which besides anorexia is also based on increased energy expenditure, relays not only on BAT thermogenesis[3,4], but also in browning-induced

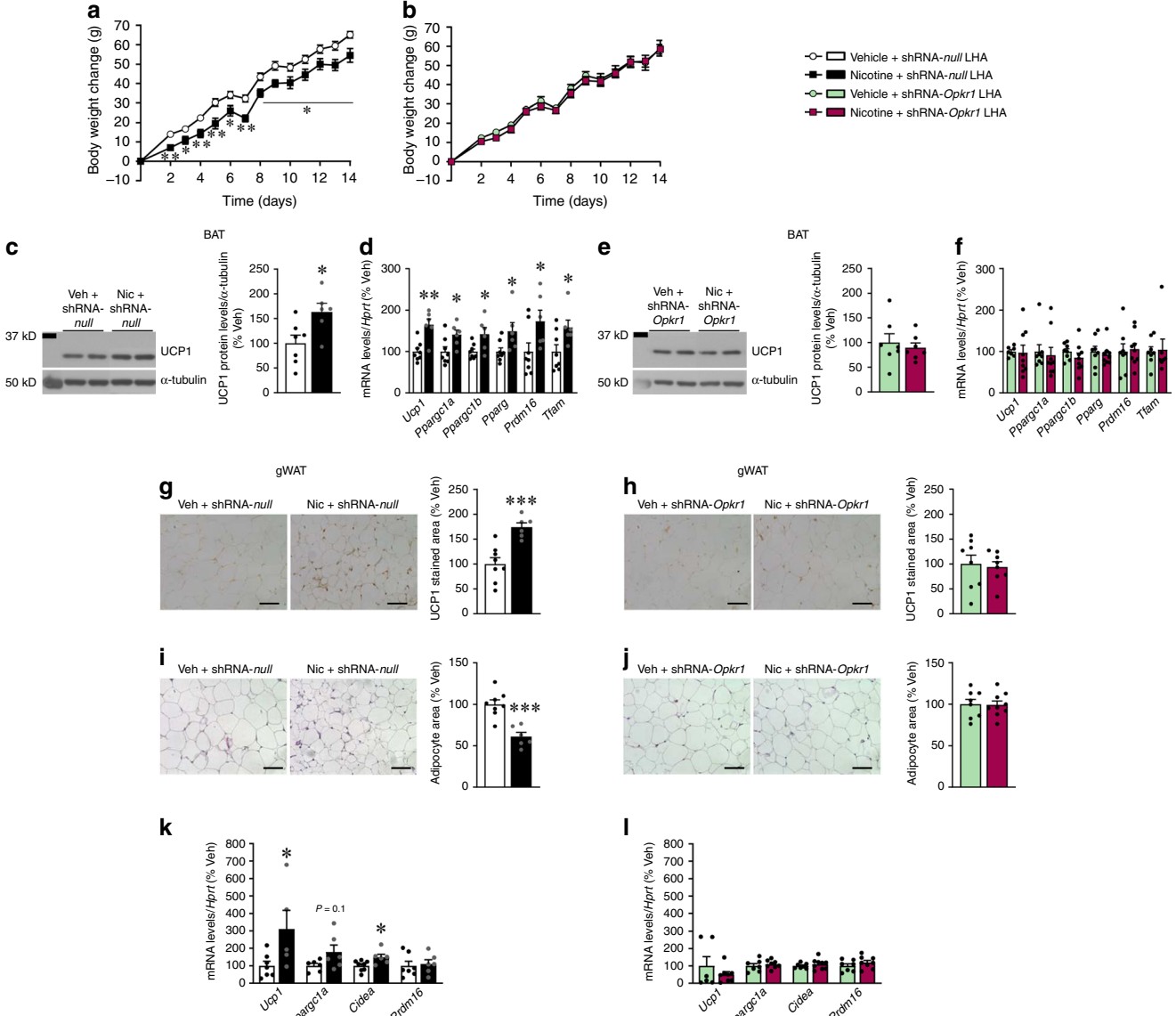

**Fig. 5** Effect of LHA *Opkr1* knockdown on nicotine's effect on BAT and WAT. **a–b** Body weight (vehicle + shRNA-*null* LHA *n* = 8, nicotine + shRNA-*null* LHA *n* = 6, vehicle + shRNA-*Opkr1* LHA *n* = 8, nicotine + shRNA-*Opkr1* LHA *n* = 9 rats). **c–e** Representative images (left panels) and protein levels of UCP1 in the BAT (right panels) (vehicle + shRNA-*null* LHA n = 7, nicotine + shRNA-*null* LHA *n* = 6, vehicle + shRNA-*Opkr1* LHA *n* = 7, nicotine + shRNA-*Opkr1* LHA *n* = 7 rats). **d–f** mRNA levels of thermogenic markers in the BAT (vehicle + shRNA-*null* LHA *n* = 8, nicotine + shRNA-*null* LHA *n* = 6, vehicle + shRNA-*Opkr1* LHA *n* = 7, 8, 8, 8, 8, 8, nicotine + shRNA-*Opkr1* LHA *n* = 9, 9, 9, 9, 9, 7 rats). **g–h** UCP1 staining (left panels 20 × , scale bar: 100 μm) and UCP1 stained area (right panels) in gWAT (vehicle + shRNA-*null* LHA *n* = 8, nicotine + shRNA-*null* LHA *n* = 6, vehicle + shRNA-*Opkr1* LHA *n* = 8, nicotine + shRNA-*Opkr1* LHA *n* = 8 rats). **i–j** H&E staining (left panels 20×, scale bar: 100 μm) and adipocyte area (right panels) in gWAT (vehicle + shRNA-*null* LHA *n* = 8, nicotine + shRNA-*null* LHA *n* = 6, vehicle + shRNA-*Opkr1* LHA *n* = 8, nicotine + shRNA-*Opkr1* LHA, *n* = 9 rats). **k–l** mRNA levels of thermogenic markers in the gWAT (vehicle + shRNA-*null* LHA *n* = 7, 6, 8, 7, nicotine + shRNA-*null* LHA *n* = 5, 6, 6, 6, vehicle + shRNA-*Opkr1* LHA *n* = 6, 7, 7, 7, nicotine + shRNA-*Opkr1* LHA *n* = 8, 8, 9, 8 rats)· of rats stereotaxically treated within the LHA with AAVs harboring a shRNA-*null* (**a**, **c**, **d**, **g**, **i** and **k**) or a shRNA against *Opkr1* (**b**, **e**, **f**, **h**, **j** and **l**) and treated with vehicle or nicotine. Center values represent average; error bars represent SEM. Statistical significance was determined by two-sided *t*-Student; **P* < 0.05, ***P* < 0.01, ****P* < 0.001 vs. vehicle. In the panels **c** and **e** showing images of gels, all the bands for each picture come from the same gel but they have been spliced for clarity. Source data are provided as a Source Data file

thermogenesis, given that UCP1 in brite/beige adipose tissue mitochondria are functionally thermogenic[42]. In relation with this, the key role of the SNS in the control of UCP1 expression and in eliciting browning[36,41], suggested us the possibility that nicotine may induce the activation of the sympathetic nerves innervating WAT. Our data showed that central nicotine not only elicited sympathetic activity and NE concentration, it also promoted changes in the sympathetic plasticity, as shown by increases dendrite density in gWAT after central nicotine administration. This is of relevance, because it demonstrates that in addition to functional alterations, central nicotine induced trophic actions on sympathetic innervation. This is a result that will deserve further investigation and it is of therapeutic interest, since it might be used to induce local increases in SNS tone that would avoid the undesired side effects that characterize overall sympathetic activation[43].

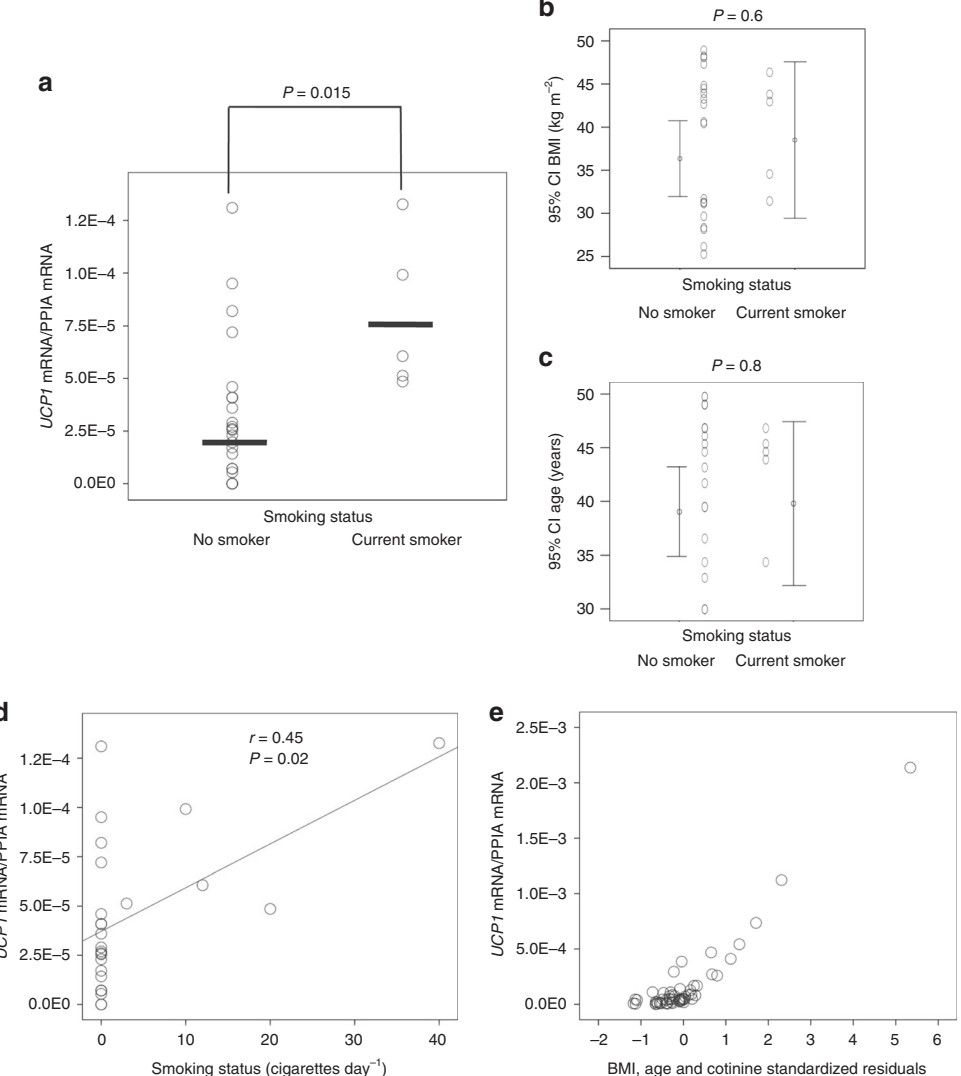

**Fig. 6** Effect of smoking status on WAT browning. **a–c** UCP1 mRNA levels in sWAT, BMI and age in non-smokers or current smokers from cohort 1 (Non smoker $n = 20$, Smoker $n = 5$). Statistical analysis was performed with one factor ANOVA with post-hoc Bonferroni test; the samples represent biological replicates. Center values (black lines) represent median. **d** Bivariate correlation between sWAT UCP1 mRNA levels and cigarettes per day on current smokers from cohort 1 ($n = 25$). This correlation was similar using parametric (Pearson, $r = 0.46$, $P = 0.02$) or non-parametric (Spearman, $r = 0.45$, $P = 0.02$) correlations. **e** Relationship between the BMI, age, and cotinine adjusted residuals of the multiple linear regression model and UCP1 gene expression in vWAT ($n = 56$). The samples represent biological replicates. Source data are provided as a Source Data file

To gain further insight on nicotine action on energy balance and specifically on WAT, we investigated the potential contribution of the opioid system[11]. We observed that global KOR deletion totally abrogated the effect of nicotine on energy homeostasis and browning. These findings led us to ask the question in which tissue(s) KOR action was relevant. Given the evidence indicating that browning can be evoked by direct action of nicotine receptors in WAT[9], we investigated the expression of KOR in adipose tissue. However, we found no expression of KOR in WAT, indicating that the effect of KOR on browning did not occur at this level. KOR is broadly expressed in the CNS, particularly the hypothalamus, where it has been involved in the control of metabolism[11,23,26]. Since within the hypothalamus, several nuclei such as the VMH and the LHA have been implicated in the control of browning[30,31,33,36,41], we selectively knocked down Opkr1 in those nuclei. Our first candidate was the VMH based on our previous report that nicotine modulates BAT thermogenesis through an AMPK/SNS-dependent mechanism

within this nucleus[3]. However, specific targeting of KOR in the VMH affected neither the catabolic action of nicotine nor its effects on BAT and WAT. This is of importance, because it suggests that AMPK in the VMH, a main regulator of browning[32,38,44], is not involved in the actions of nicotine on WAT. Next, we examined whether KOR in the LHA could be involved in the effect of nicotine. Recent data reported that AMPK in the VMH controls LHA function (through a glutamatergic-dependent mechanism) to regulate energy homeostasis[33]. Remarkably, when KOR was ablated in the LHA the effect of nicotine on any aspect of energy balance was totally abolished. Thus, the LHA KOR is critical for the catabolic effect of central nicotine and its actions on browning. At physiological level, it is conceivable that cholinergic neurons in both the LHA (or other hypothalamic nuclei) and in the periphery may modulate the activity of white adipocytes and, subsequently the process of browning. However, the relevance of the peripheral mechanism associated with the induction of browning in human

smokers is, in our view, unlikely making the central effect more plausible. This is based on the fact that the brain is the main target of nicotine[34,35,45]. In fact, while circulating and peripheral levels of nicotine are highly variable, its levels are higher and its fluctuations much smaller in the brain[45].

In summary, our results show that besides its possible direct action on nicotinic receptors in WAT, nicotine modulates browning by acting centrally and modulating the sympathetic firing on white fat. Moreover, we demonstrate that an intact KOR in the LHA is critical for nicotine to exert its catabolic and browning-inducing actions. Overall, this evidence suggests that the functional dependence of nicotine on KOR may offer new possible targets for the treatment of obesity and associated comorbidities.

## Methods

**Animals**. Adult male Sprague-Dawley rats (8–10-weeks-old, 200–250 g; Animalario General USC, Santiago de Compostela, Spain) and adult male null κ opioid receptor (KOR KO) mice and their wildtype controls (8–12-weeks-old, 20–25 g)[23] were used. The animals were housed with a 12-h light (8:00 to 20:00)/12-h dark cycle, under controlled temperature and humidity conditions and allowed free access to standard laboratory chow (STD, SAFE A04: 3,1% fat, 59,9% carbohydrates, 16,1% proteins, 2.791 kcal g$^{-1}$; Scientific Animal Food and Engineering; Nantes, France) and tap water. All experiments were performed in agreement with the International Law on Animal Experimentation and were approved by the USC Ethical Committee (Project ID 15010/14/006), Gulbenkian Ethics Committee (Project ID A011.2016) and the University of Iowa Institutional Animal Care and Use Committee.

**Patients**. Two independent cohorts (cohort 1, $n = 25$ and cohort 2, $n = 56$) were studied according to smoking status. Anthropometric (sex, age, and BMI) and clinical (circulating levels of glucose, total cholesterol, HDL cholesterol, LDL cholesterol and triglycerides, as well as cotinine) parameters from these participants are detailed in Supplementary Table 1. All these subjects were recruited at the Endocrinology Service of the Hospital of Girona Dr. Josep Trueta, were of Caucasian origin and reported that their body weight had been stable for at least three months before the study. Subjects were studied in the post-absorptive state. BMI was calculated as weight (in kg) divided by height (in m) squared. Patients had no systemic disease other than obesity and all were free of any infections in the previous month before the study. Liver diseases (specifically tumoral disease and HCV infection) and thyroid dysfunction were specifically excluded by biochemical work-up. All subjects gave written informed consent, validated and approved by the Ethical Committee and the Committee for Clinical Investigation of the Hospital Universitari Dr. Josep Trueta (Girona, Spain), after the purpose of the study was explained to them. Samples and data from patients included in this study were partially provided by the FATBANK platform promoted by the CIBEROBN and coordinated by the IDIBGI Biobank (Biobank IDIBGI, B.0000872), integrated in the Spanish National Biobanks Network and they were processed following standard operating procedures with the appropriate approval from the patient (informed consent) and also of the Ethics, External Scientific and FATBANK Internal Scientific Committees. We certify that all applicable institutional regulations concerning the ethical use of information and samples from human volunteers were followed during this research. Adipose tissue samples were obtained during elective surgical procedures (cholecystectomy, surgery of abdominal hernia and gastric by-pass surgery). Samples of WAT were immediately transported to the laboratory (5–10 min). The handling of tissue was carried out under strictly aseptic conditions. WAT samples were washed in PBS, cut off with forceps and scalpel into small pieces (100 mg), and immediately flash-frozen in liquid nitrogen before being stored at −80 °C. Serum cotinine was measured by cotinine ELISA kit (KA0930, Abnova; Taipei City, Taiwan) following manufacturer' instructions.

**Treatments**. Rats were treated SC with nicotine (nicotine-hydrogen-tartrate salt, 2 mg kg$^{-1}$ every 12 h; Sigma; St Louis, MO, USA)[3,4]. Mice were house individually and implanted subcutaneously with an osmotic minipump flow moderator (Model 1002, Alzet, DURECT Corporation; Cupertino, CA, USA) containing a daily dose of 25 mg kg$^{-1}$ of nicotine (nicotine-hydrogen-tartrate salt, Sigma; St Louis, MO, USA)[46] due to their higher metabolism, or saline. Body weight was measured for 14 days. For the central nicotine experiments, intracerebroventricular (ICV) cannulae were stereotaxically implanted under ketamine/xylazine anesthesia[3,33,47–50]. Animals were individually housed and used for experimentation four days later. Then, animals were subjected to a protocol of daily ICV injections of nicotine (nicotine-hydrogen-tartrate salt, 0.3 μg day$^{-1}$; Sigma; St Louis, MO, USA)[51] or vehicle (3 μl of saline; control mice) using a 22-gauge needle (Hamilton; Reno, NV, USA) through the inserted cannulae for 7 days.

**Stereotaxic microinjection of viral vectors**. We used adult male Sprague-Dawley rats (200 g, Animalario General, USC). Rats were placed in a stereotaxic frame (David Kopf Instruments; Tujunga; CA, USA) under ketamine-xylazine anesthesia. The hypothalamic nuclei were targeted bilaterally using a 25-gauge needle (Hamilton; Reno, NV, USA). The injections were directed to the following stereotaxic coordinates: (a) for the VMH: 2.4/3.2 mm posterior to the bregma (two injections were performed in each VMH), ±0.6 mm lateral to midline and 10.1 mm ventral[3,33,38,47–49] and (b) for the LHA: 2.85 mm posterior to the bregma, ±2 mm lateral to the midline and 8.1 mm ventral[24,33]. Adeno-associated viral (AAV) vectors ($1 \times 10^9$ genomic copies μl$^{-1}$) encoding or not rat KOR short-hairpin RNAs[23] were delivered at a rate of 200 nl min$^{-1}$ for 5 min (1 μl at each injection site) to specifically silence the expression of *Opkr1* mRNA in the VMH or the LHA. After 14 days rats were implanted subcutaneously with an osmotic minipump flow moderator (Model 2002, Alzet, DURECT Corporation, Cupertino, CA, USA) containing a daily dose of 4 mg kg$^{-1}$ of nicotine (nicotine-hydrogen-tartrate salt, Sigma; St Louis, MO, USA) or saline.

**Indirect calorimetry**. Animals were analyzed for EE, respiratory quotient (RQ) and locomotor activity (LA) using a calorimetric system (*LabMaster; TSE Systems*; Bad Homburg, Germany)[3,33,48,49]. Animals were placed in a temperature-controlled (24 °C) box through which air was pumped. After calibrating the system with the reference gases (20.9% O2, 0.05% CO2, and 79.05% N2), the metabolic rate was measured for 24–48 h. EE, RQ (VCO2/VO2) and LA were recorded every 30 min. Animals were placed for adaptation for 1 week before starting the measurements.

**Nuclear magnetic resonance**. For the measurement of body composition, we used nuclear magnetic resonance (NMR) (Whole Body Composition Analyzer; EchoMRI; Houston, TX and Bruker BioSpin; Ettlingen, Germany)[33,49,52,53].

**Sympathetic nerve activity recording**. For multi-fiber recording of sympathetic nerve activity (SNA) C57BL/6 J mice were anesthetized using intraperitoneal ketamine (91 mg kg$^{-1}$) and xylazine (9.1 mg kg$^{-1}$) and anesthesia maintained with α-chloralose (initial dose: 12 mg kg$^{-1}$, sustaining dose: 6 mg kg$^{-1}$ each hour) via a catheter inserted in the right jugular vein. The multi-fiber recording of SNA was obtained from nerve fibers subserving the sWAT located in the inguinal region of the right hindlimb. Using a dissecting microscope, a nerve fiber innervating sWAT was identified, placed on a bipolar platinum-iridium electrode. Each electrode was attached to a high-impedance probe (HIP-511, Grass Instruments; Warwick, RI, USA) and the nerve signal was amplified $10^5$ times with a Grass P5 AC pre-amplifier (Grass Instruments; Warwick, RI, USA). After amplification, the nerve signal was filtered at a 100-Hz and 1000-Hz cutoff with a nerve traffic analysis system (Model 706C, University of Iowa Bioengineering; Iowa City, IA, USA). The nerve signal was then routed to an oscilloscope (Model 54501A, Hewlett-Packard; Palo Alto, CA, USA) for monitoring the visual quality of the sympathetic nerve recording and to a resetting voltage integrator (Model B600c, University of Iowa Bioengineering; Iowa City, IA, USA). Basal sWAT SNA measurements were made during a 10 min control period. sWAT SNA was continuously measured throughout the next 4 h after ICV injection of either nicotine or vehicle. To ensure that electrical noise was excluded in the assessment of sympathetic outflow, we corrected each SNA recording for post-mortem background activity[47–49,52,54].

**Sample processing**. Rats and mice were killed by cervical dislocation. From each animal, the hypothalamus (or VMH and LHA), striatum, cortex, BAT; gonadal, subcutaneous inguinal and visceral WAT were harvested and immediately homogenized on ice to preserve phosphorylated protein levels. Samples were stored at −80 °C until further processing. For immunohistochemistry and histology analyses samples were fixed in formalin 10% and lately paraffin embedded.

**NE measurement**. To assess the content of NE in innervated WAT, 12-week-old WT mice were killed for tissue collection to examine the effect of nicotine ICV treatment. Gonadal, subcutaneous inguinal and visceral fat pads were collected for homogenization in buffer (1 N HCl, 1 mM EDTA, 4 mM Sodium metabisulfire). NE levels were determined with a NE ELISA kit (Labor Diagnostika Nord GmbH and CoKG; Nordhorn, Germany)[55]. Tissues were homogenized and sonicated and cellular debris was pelleted by centrifugation at 16,000×$g$ for 15 min at 4 °C. All tissue samples were normalized to total tissue protein concentration.

**High-performance liquid chromatography (HPLC)**. In WT and KOR null mice, the hypothalamus, prefrontal cortex and striatum were dissected on an ice-cold plate, and immediately frozen on dry ice and stored at −80 °C until analysis. Then tissue was homogenized and then centrifuged (14,000×$g$; 10 min at 4 °C). The remaining supernatant fraction was filtered and injected (20 μl/injection) into the High-Performance Liquid Chromatography (HPLC) system (Shimadzu LC Prominence; Shimadzu Corporation; Kyoto, Japan)[56,57]. Dopamine and its metabolite, DOPAC, 5-HT and NE were separated using a reverse phase analytical column (Waters Symmetry 300C18; Waters, Milford, MA, US). The mobile phase consisted of a 10% MeOH solution (pH = 4) containing 70 mM KH2PO4, 1 mM

octanesulfonic acid and 1 mM EDTA, and was delivered at a rate of 1 ml min$^{-1}$. Detection was performed with a coulometric electrochemical detector (ESA Coulochem III; ESA, Chelmsford, MA). Data were processed with the Shimadzu LC Solution Software (Shimadzu Corporation; Kyoto, Japan) and were expressed as ng mg$^{-1}$ of wet tissue.

**Real-time quantitative RT-PCR**. Real-time PCR (TaqMan®; Applied Biosystems; Foster City, CA, USA) was performed using specific primers and probes (Supplementary Table 3)[33,47–49,54]. Values were expressed in relation to hypoxanthine-guanine phosphoribosyl-transferase (Hprt) levels. For the analysis of the human WAT samples, we used commercially available and pre-validated TaqMan® primer/probe sets (Applied Biosystems; Carlsbad, CA, USA) as follows: endogenous control peptidylprolyl isomerase A (cyclophilin A) (PPIA, 4333763) and UCP1[38]. Gene expression values were expressed relative to PPIA levels.

**Immunohistochemistry**. Adipose tissue depots were fixed in 10% buffered formaldehyde and paraffin embedded. For the hematoxylin–eosin processing, the WAT sections were first stained with hematoxylin for 5 min, washed and stained again with eosin for 1 min. Detection of uncoupling protein 1 (UCP1) in WAT was performed using anti-UCP1 antibody (1:500; ab10983; Abcam, Cambridge, UK)[38,41,44]. Images were taken with a digital camera Olympus XC50 (Olympus Corporation; Tokyo, Japan) at 20× for rats and 40× for mice. Digital images for WAT were quantified with ImageJ 1.44 Software (National Institutes of Health; USA)[38,41,44].

**Adipo-Clear**. Fluorescence-imaging of the fat pads was based on current protocols[28,29] with some modifications. Mice were sacrificed with $CO_2$ and perfused with 1× PBS. The gWAT was harvested, the attached connective tissues were removed under a dissecting microscope and finally the samples were fixed in 4% PFA at 4 °C overnight. Fixed fat pads were washed in PBS for 15 min three times at RT. Fixed samples were washed in 20%, 40%, 60%, 80%, 100%, 100% methanol in $H_2O$ for 30 min each shaking at RT. Then a bleaching treatment was performed with 5% $H_2O_2$ in methanol for 24 h at 4 °C shaking. After that, samples were rehydrated with methanol 80%, 60%, 40%, 20% for 30 min each at RT and washed twice with PBS/0.2% Tween-20, 10 µg ml$^{-1}$ heparin for another 30 min each at RT shacking. The samples were permeabilized for 16 h at 37 °C shaking in a perm solution (20% DMSO, 0.2% TritonX-100, 0.3 M glycine in PBS). The tissues were blocked with 10% DMSO, 5% goat serum, 0.2% TritonX-100 in PBS for 12 h at 37 °C shaking. The tissues were immunolabeled with TH primary antibody (1:500; ab152; Merk Millipore; Burlington, MA, US) diluted in 5% DMSO, 5% goat serum, 0.2% Tween-20, 10 µg ml$^{-1}$ Heparin in PBS for 3 days at 37 °C shaking and then they were washed with 1× PBS, 0.2% Tween-20, 10 µg ml$^{-1}$ Heparin for 1 h seven times at RT shaking with a last wash performed overnight. The fat pads were subsequently immunolabeled with Alexa Flour 594 secondary antibody (1:500; A11042; Life Technologies; Carlsbad, CA, US) diluted in 5% DMSO, 5% goat serum, 0.2% Tween-20, 10 µg ml$^{-1}$ Heparin in PBS for 2 days at 37 °C shaking and were washed with 1× PBS, 0.2% Tween-20, 10 µg ml$^{-1}$Heparin for 1 h seven times at RT shaking, with a final wash overnight. The immunolabeled adipose tissues were embedded in 1% agarose-blocks (UltraPure agarose; Invitrogen; Carlsbad, CA, US) in PBS. The blocks were dehydrated with 20%, 40%, 60%, 80%, 100% methanol in $H_2O$ for 1 h each at RT with shaking and 100% methanol overnight. The blocks were incubated in a solution 1:1 of methanol-BABB (benzyl alcohol and benzyl benzoate, 1:2) for 4 h shaking at RT and then with BABB solution until the block cleared (24 h). The optically cleared adipose tissues were imaged by optical projection tomography (OPT) using a 0,77× lens. The full series of projections of the whole fat pad were acquired from multiple angles (1600 angles) and then pre-processed for back-projection reconstruction using FIJI and the axis of rotation re-aligned using Skyscan's NRecon Software[58]. 3D reconstruction and quantification were performed with IMARIS 9.2 Software (Bitplane; Zurich, Switzerland). Sympathetic Density Quantification were performed with the Filament Tracer tool[28]. Briefly, in each fat pad, we randomly isolated small cubic areas (7–8 cubes) contained within lobules. TH signal was reconstructed with the Filament Tracer tool and the regional volumes (µm$^3$), areas (µm$^2$), and total neurite length (µm) of the cubes were calculated.

**Western blotting**. Protein lysates from the VMH, LHA and BAT were subjected to SDS-PAGE, electrotransferred to polyvinylidene difluoride membranes (PVDF; Merck Millipore; Billerica, MA, USA) with a semidry blotter and probed with antibodies against UCP1 (1:10,000; ab10983), MOR (1:1000; ab17934), β2-nAChR (1:5000; ab41174), β4-nAChR (1:1000; ab129276), α3-nAChR (1:1000; ab183097), α4-nAChR (1:1000; ab41172), α7-nAChR (1:1000; ab216485) (Abcam; Cambridge, UK); β-actin (1:5000; A5316), α-tubulin (1:5000; T5168), KOR (1:1000; SAB2501442), DOR (1:1000; SAB4502042) (Sigma; St Louis, MO, USA); AMPKα1 (1:1000; 07–350), AMPKα2 (1:1000; 07–363) (Millipore; Billerica, MA, USA), pAMPKα-Thr$^{172}$ (1:1000; 2535S) (Cell Signaling; Danvers; MA, USA)[3,48–50,54]. Autoradiographic films (Fujifilm, Tokyo, Japan) were scanned and the bands signal was quantified by densitometry using ImageJ-1.44 software (NIH; Bethesda, MD, USA)[3,48–50,54]. Values were expressed in relation to β-actin (hypothalamus) or α-tubulin (BAT). Representative images for all proteins are shown; all the bands for each picture come always from the same gel, although they may be spliced for

clarity. Uncropped and unprocessed scans of the showed blots are supplied in the Source Data file.

**Statistical analysis**. In animal experiments, statistical significance was determined by two-sided t-Student. $P < 0.05$ was considered significant; error bars represent SEM. In human studies, statistical analyses were performed using SPSS 12.0 Software. Unless otherwise stated, descriptive results of continuous variables are expressed as mean and SD for Gaussian variables or median and interquartile range. The relation between variables was analyzed by simple correlation (Pearson's test and Spearman's test) and multiple linear regression analysis. One factor ANOVA with post-hoc Bonferroni test were used to compare no smoker vs. current smoker group. Levels of statistical significance were set at $P < 0.05$.

**Reporting summary**. Further information on research design is available in the Nature Research Reporting Summary linked to this article.

## Data availability

The data that support the findings of this study are available from the corresponding author upon reasonable request. The source data underlying Figures and Supplementary Figures are provided as a Source Data file.

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

## Acknowledgements

We would like to thank the technical support of IGC's Advanced Imaging Facility (AIF-UIC) and of Micron Oxford bioimaging unit. The research leading to these results has received funding from the Xunta de Galicia (J.L.L.-G.: ED431C 2018/10; R.N.: 2015-CP080 and 2016-PG057; M.L.: 2015-CP079 and 2016-PG068); Ministerio de Economía y Competitividad (MINECO) co-funded by the FEDER Program of EU (J.L.L.-G.: BFU2015–70523; R.N.: BFU2015–70664R; C.D.: BFU2017–87721-P; M.L.: RTI2018-101840-B100; BFU2015–70454-REDT/Adipoplast and RTI2018–101840-B-I00); Instituto de Salud Carlos III (J.L.L.-G.: RD16/0011/0016; J.M.F.-R.: PI15–01934); European Molecular Biology Organization (A.D.: EMBO-Installation Grant 3037); Human Frontier Science Program (A.D.: HFSP-RGY0070/2016); Howard Hughes Medical Institute (A.D.: HHMI-208576/Z/17/Z); US National Institutes of Health (K.R.: HL084207); American Heart Association (K.R.: EIA#14EIA18860041); the University of Iowa Fraternal Order of Eagles Diabetes Research Center (K.R.); Atresmedia Corporación (R.N. and M.L.: 2017-PO004); Fundación BBVA (R.N.), European Foundation for the Study of Diabetes (R.N.); and ERC Synergy Grant-2019-WATCH-810331 (R.N.). P.S.-C. is recipient of a fellowship from Xunta de Galicia (ED481B 2018/050). L.L.-P. is recipient of a fellowship from Xunta de Galicia (ED481A-2016/094); E.R.-P. is recipient of a fellowship from MINECO (BES-2015–072743). N.M.-S. is recipient of a fellowship from Xunta de Galicia (ED481B 2016/168–0) and from the European Union's Horizon 2020 research and innovation programme under the Marie Sklodowska-Curie actions. The CiMUS is supported by the Xunta de Galicia (2016–2019, ED431G/05). *CIBER* de Fisiopatología de la Obesidad y Nutrición is an initiative of ISCIII. The funders had no role in study design, data collection and analysis, decision to publish, or preparation of the paper. We also want to particularly acknowledge the patients, the *FATBANK* platform promoted by the *CIBEROBN* and the *IDIBGI* Biobank (Biobanc *IDIBGI*, B.0000872), integrated in the Spanish National Biobanks Network, for their collaboration and coordination.

## Author contributions

P.S.-C., L.L.-P., E.R.-P., A.R.-P. A.S., N.M.-S. and C.F. performed the in vivo experiments, analytical methods (indirect calorimetry, NRM, NE measurements, RT-PCR, immunohistochemistry, and western blotting), collected and analyzed the data. J.M.M.-N. and J.M.F.-R. performed the human studies. P.G.-G. and J.L.L.-G. performed the HPLC studies. R.I. and T.S. performed the NMR studies. D.A.M. and K.R. performed and analyzed the sympathetic nerve activity data. N.M.-S., N.T., S.M., and A.D. performed the Adipo-Clear analysis. P.S.-C., G.M.-G., J.L.L.-G, R.N., A.D, J.M.F.-R., K.R., C.D. and M.L. conceived and designed the experiments, analyzed, interpreted and discussed the data. P.S.-C. and M.L. made the figures. M.L. wrote the paper (which all authors reviewed and edited), developed the hypothesis, secured funding, coordinated, and led the project. J.L.L.-G., R.N., A.D., J.M.F.-R. and K.R. contributed equally.

## Additional information

**Competing interests:** The authors declare no competing interests.

