## [Peer Review File · Nature Communications]

Reviewers' comments:

Reviewer #2 (Remarks to the Author):

The paper by Seoane-Collazo et al. investigates the browning effect of nicotine in rodent and human white adipose tissue depots. They report that nicotine, by acting at central level through a kappa opioid receptor-mediated neurocircuit in the lateral hypothalamus, determines an increased sympathetic tone to peripheral white fat depots leading to appearance/increase of brown fat into white, or predominantly white, adipose depots (browning). This hypothesis is novel and potentially important both for smoke addiction studies and for understanding the central mechanisms of food intake and energy expenditure regulation and obesity. In my opinion, however, methods are not adequate and further experimental evidence is required to strengthen the conclusions.

Major points

In the Methods section 1) which adipose depots have been analyzed in the study are not reported (!). Throughout the paper, it seems to me that which adipose depot(s) were examined is never specified! This is instead very important because not all adipose depots exhibit similar browning abilities ... 2) Morphometric analyses should be specified. 3) Stereotaxic details for PVN are not given. 4) Nerve recording experiments should be better detailed (again ... which nerves from which depot?). 5) It is not specified whether immunohistochemistry for UCP1 was performed on fixed and paraffin embedded tissues, or frozen specimens.

The Authors claim that the browning abilities of central administered nicotine is due to an increased sympathetic tone to white fat. This assumption mainly comes from electrophysiological experiments. Given that this is a crucial point of the study I think that this assumption should be corroborated by other experimental approaches including dosage of noradrenaline in white fat from treated and not-treated mice and/or quantitative assessment of noradrenergic innervation in the adipose tissues by tyrosine hydroxylase western blotting and/or immunohistochemistry.

Minor points

Introduction, line 6: involves not "involve";

Not all readers are familiar with "cotinine": please, specify.

Reviewer #3 (Remarks to the Author):

The article from Seoane-Collazo et al. investigated the effects of nicotine on the browning of white fat through kappa opioid receptor signaling, and how this potential mechanism could contribute to nicotine-induced weight loss, mainly through thermogenesis. These experiments are also complemented by relevant clinical data. This is an elegant and comprehensive study providing a novel molecular mechanism underlying the increase in calorie expenditure induced by nicotine. The experiments are adequately designed and controlled, and the rationale for each sub-aim is well-defined. Clinical data is limited but still add a relevant opening for future directions. The figures are clear and properly described. I only have minor suggestions highlighted below:

INTRODUCTION

CHRNA: it may be preferable to change this abbreviation as it is a gene name.

RESULTS

Is it known whether KOR mutants have nicotinic receptor abnormalities that could underlie the observed effects?

Are KOR mutants different from wild type animals at baseline regarding their fat depot and metabolism? Because energy expenditure is not a linear variable, this could also underlie some of the observed effects or lack thereof.

METHODS

There is the assumption that mice and rats have similar (fat) metabolism and reactivity to nicotine treatment. Because species differences have clearly been emphasized, please provide more justifications to alternate between both species.

Overall, the n is relatively limited in many of the animal groups, but not all, which seems curious as the same animals should have been used for several consecutive experiments reported in this article. Were some of the samples a pool of several animals? If so, please specify.

DISCUSSION

The discussion is very limited but so is space, although some of the text could be shortened as it is only summarizing the results. Both in the introduction and the discussion, $\alpha 2$ nAChR is mentioned but since none of the experiments investigated the effects of $\alpha 2$ modulation, the discussion around this receptor is largely irrelevant. Instead, it would be preferable to extend the discussion on UCP-1.

REVIEWER#2

Overall comment: *The paper by Seoane-Collazo et al. investigates the browning effect of nicotine in rodent and human white adipose tissue depots. They report that nicotine, by acting at central level through a kappa opioid receptor-mediated neurocircuit in the lateral hypothalamus, determines an increased sympathetic tone to peripheral white fat depots leading to appearance/increase of brown fat into white, or predominantly white, adipose depots (browning). This hypothesis is novel and potentially important both for smoke addiction studies and for understanding the central mechanisms of food intake and energy expenditure regulation and obesity. In my opinion, however, methods are not adequate and further experimental evidence is required to strengthen the conclusions.*

Response: We thank the Reviewer for the positive view of our manuscript. We also believe that the current manuscript provides novel and important data that enhances our understanding of the central role of nicotine in body metabolism regulation. A detailed point-by-point response to the comments is included below. As the Reviewer will note, following his/her advice we have performed new experimental settings that improve our former observation and strengthen the main conclusions of the manuscript.

Major points

Reviewer's comment 1: *In the Methods section 1) which adipose depots have been analyzed in the study are not reported (!). Throughout the paper, it seems to me that which adipose depot(s) were examined is never specified! This is instead very important because not all adipose depots exhibit similar browning abilities*

Response: This is a very important point that we further address in this revised manuscript. The original analyses had been performed in gonadal white adipose tissue (gWAT), in which browning is well known to be under CNS control¹⁻¹⁰. However, as part of our routine dissection protocols, we also kept sWAT and vWAT. Therefore, following the suggestion of Reviewer we have analyzed the effect of browning in the sWAT and vWAT in the following settings, which we agreed with the Editor: **1)** ICV injection of nicotine (**Figure 1g-j**) and **2)** SC injection of nicotine (**Suppl. Figure 1i-l**). The new data demonstrate that nicotine, either given centrally or peripherally induced the browning on sWAT and vWAT depots in a similar extent to that observed in gWAT (**Figure 1d-f and Suppl. Figure 1g-h**). In keeping with this evidence, our data also show (please, see response to **Reviewer#2's comment 6**) that central nicotine increases (or trends to increase) the NE levels in all the evaluated WAT depots (**Figure 2c**), as well as the sympathetic innervation of gWAT (**Figure 2d**). Overall, this evidence strengthens the idea that nicotine, by acting centrally, induces the WAT browning through a sympathetic mechanism.

Reviewer's comment 2: *Morphometric analyses should be specified*

Response: We agree with this point, as we assume the Reviewer means the human data. This information was already presented in the former version of the manuscript (**Suppl.**

Table 1). However, following his/her advice this has been better specified in the new version.

Reviewer's comment 3: Stereotaxic details for PVN are not given.

Response: We apologize for not giving these details. This information has been added in the revised version.

Reviewer's comment 4: Nerve recording experiments should be better detailed (again ... which nerves from which depot?)

Response: We apologize for being brief in describing the details of this protocol. This information has been explained in deep in the revised version.

Reviewer's comment 5: It is not specified whether immunohistochemistry for UCP1 was performed on fixed and paraffin embedded tissues, or frozen specimens.

Response: The Reviewer is right: this information was not given in the former version. UCP1 IHC was performed on fixed and paraffin embedded tissues.

Reviewer's comment 6: The Authors claim that the browning abilities of central administered nicotine is due to an increased sympathetic tone to white fat. This assumption mainly comes from electrophysiological experiments. Given that this is a crucial point of the study I think that this assumption should be corroborated by other experimental approaches including dosage of noradrenaline in white fat from treated and not-treated mice and/or quantitative assessment of noradrenergic innervation in the adipose tissues by tyrosine hydroxylase western blotting and/or immunohistochemistry.

Response: This is a very interesting point and we thank the Reviewer for it. We start working with nicotine about 9-10 years ago. In that particular set of experiments, there was a specific result that focused our attention: the fact that the weight-reducing effect of peripheral (SC) nicotine administration was partially blunted by a β 3-AR antagonist (SR59230A¹⁰⁻¹⁵; also SC given; **Rebuttal Figure 1a**), which was indicative that the effect of nicotine on energy balance was dependent on the SNS. Notably, that action affected only body weight but not to feeding (**Rebuttal Figure 1b**), suggesting that the expenditure side of the energy balance equation was being affected; this was further demonstrated by the normalization in the UCP1 mRNA levels in BAT after SR59230A co-treatment (**Rebuttal Figure 1c**). Therefore, this led us to investigate the effect of nicotine on brown adipose tissue (BAT) thermogenesis^{16,17}. Furthermore, considering that the hypothalamus orchestrates the sympathetic control of BAT thermogenesis and importantly WAT browning¹⁸⁻²⁰, these data prompted us to address the central effects of nicotine. In the first version of the manuscript, we used only an electrophysiological approach because, in our view, it provided a straight and functional readout. In this new version, for a better characterization of the SNS involvement and following the advice of the Reviewer, we have better analyzed the effect of central nicotine on WAT browning, in collaboration with our colleague and collaborator Prof. Ana Domingos (*Oxford University; UK*), who is a worldwide recognized expert in the SNS innervation of white fat^{21,22}. Firstly, we analyzed the NE content in the different WAT depots after central nicotine administration. Our data showed that central nicotine induced an increase in the NE concentration in all the analyzed depots (**Figure 2c**). Secondly, we investigated whether nicotine might affect the sympathetic innervation of gWAT. For this purpose, we have used *Adipo-Clear*, a whole-tissue clearing method that permits immunolabeling and three-dimensional profiling, to identify differences in tissue architecture and sympathetic innervation in adipose depots^{23,24}. Using a novel approach, the new set of data showed that central nicotine trends to increase the density of sympathetic innervation in gWAT (**Figure 2d**). Overall, this evidence strengthens the idea that nicotine, by acting centrally, induces the WAT browning through a sympathetic mechanism.

Rebuttal Figure 1: Effect of adrenergic blockage on the effect of nicotine on energy balance. (a) Body weight change; (b) daily food intake and (c) UCP1 mRNA levels in the BAT of rats injected SC with nicotine and the β 3-AR antagonist SR59230A (n=8-16 animals/group). Data expressed as mean \pm SEM. *, **, *** P<0.05, 0.01 and 0.001 vs. vehicle; # P<0.05 nicotine vs. nicotine + SR59230A. For graph simplification the SR59230A alone group was not represented, although this treatment did not induce changes in any of the analyzed parameters.

Minor points

Introduction, line 6: involves not "involve";

Not all readers are familiar with "cotinine": please, specify.

Response: These mistakes have been corrected in the new version.

We thank again the Reviewer#2 for his/her excellent insight that has improved the scope and quality of our manuscript.

REVIEWER#3

Overall comment: The article from Seoane-Collazo et al. investigated the effects of nicotine on the browning of white fat through kappa opioid receptor signaling, and how this potential mechanism could contribute to nicotine-induced weight loss, mainly through thermogenesis. These experiments are also complemented by relevant clinical data. This is an elegant and comprehensive study providing a novel

molecular mechanism underlying the increase in calorie expenditure induced by nicotine. The experiments are adequately designed and controlled, and the rationale for each sub-aim is well-defined. Clinical data is limited but still add a relevant opening for future directions. The figures are clear and properly described. I only have minor suggestions highlighted below:

Response: We thank the Reviewer for the positive assessment of our manuscript. We also believe that the current manuscript provides novel and important data that enhance our understanding of the central role of nicotine in body metabolism regulation. A detailed point-by-point response to all his/her comments is included below.

Introduction

Reviewer's comment 1: CHRNA: it may be preferable to change this abbreviation as it is a gene name.

Response: Following the advice of the Reviewer, this has been corrected in the new version. We apologize for this mistake.

Results

Reviewer's comment 2: Is it known whether KOR mutants have nicotinic receptor abnormalities that could underlie the observed effects?

Response: We have extensively checked the literature and we have not found any reported difference in this regard. In any case, following the Reviewer suggestion and to provide experimental insight on this issue, we have extensively analyzed the nicotinic receptors in the hypothalamus of KOR KO mice. Our data show that none of nicotinic receptors analyzed (β 2-nAChR, β 4-nAChR, α 2-nAChR, α 4-nAChR, α 7-nAChR) show major changes in the protein levels between WT and KOR KO mice (**Suppl. Figure 2a**), suggesting the lack of abnormalities in the mutant animals. To further explore the brain function in KOR KO mice, and in collaboration with Prof. José Luis Labandeira-García (USC), a worldwide expert in neurotransmission²⁵⁻²⁸, we have analyzed the concentration of dopamine (DA), 3,4-dihydroxyphenylacetic acid (DOPAC), serotonin (5-HT) and NE in the hypothalamus, striatum and cortex of WT and KOR KO mice to search for a major alteration at baseline in neurotransmission. Our data show similar values for all the analyzed parameters (**Suppl. Figure 2b**). Overall, this evidence suggest that the observed changes are not related to any generalized defect/-s that might characterize KOR mutant mice.

Reviewer's comment 3: Are KOR mutants different from wild type animals at baseline regarding their fat depot and metabolism? Because energy expenditure is not a linear variable, this could also underlie some of the observed effects or lack thereof.

Response: This is a very interesting point that has been already addressed in the literature. It has been reported that KOR KO mice have lower body weight and adiposity than WT mice. This effect was associated with unchanged food intake but higher energy expenditure²⁹. This is shown in **Figures 3g** (WT) and **3h** (KOR KO) of the current (former Figures 2g-h) version, where basal EE levels for the null animals are elevated. In relation of the lack of linearity, we agree about this intrinsic characteristic of EE. However, we have done the analyses at several time periods (every 30 min for 48 hours) and they are consistent. In fact, KOR mice treated with nicotine have a slight tendency to lower EE in the light phase (**Figure 3h**), which would be opposite to what was found in WT. Therefore, we believe that it is clear that nicotine does not induce EE in KOR KO mice, which in our view is in agreement with the lack of BAT and browning responses.

Methods

Reviewer's comment 4: There is the assumption that mice and rats have similar (fat) metabolism and reactivity to nicotine treatment. Because species differences have clearly been emphasized, please provide more justifications to alternate between both species.

Response: We totally agree this point, but we think that it is justified, as follows. The main reason for using rats in the stereotaxic experiments (virogenetic manipulations) relates to larger size of their brain, that makes the procedure easier. As the Reviewer may know, stereotaxic approaches in mice are often more complicated due to the small size of the brain, which implies lower efficiency of the targeting, lower reproducibility and the requirement of using many mice. This is the main reason why normally we have combined in our recent studies the utilization of mutant mice with stereotaxic settings in rats^{10,13-15,30}. In any case, in our humble opinion, the results in rats (and in humans) are positive for the main conclusion of the study. The fact that the effect of nicotine on browning is reproducible in three different species (mouse, rat and human) strengthens the conclusions instead of being a weakness.

Reviewer's comment 5: Overall, the n is relatively limited in many of the animal groups, but not all, which seems curious as the same animals should have been used for several consecutive experiments reported in this article. Were some of the samples a pool of several animals? If so, please specify.

Response: We partially agree with this point. It is true that in some of the analyses of the experiments using KOR KO mice the n is low, but this was due to the inherent difficulties of the experimental approach. For example, the effect of nicotine on the KOR (and WT) model was analyzed in two different experiments (6-7 animals/group for each experiment). Thus, the body weight data in **Figure 3a** (former Figure 2a) are the pooled data for both experiments (n=12-14); however, the rest of the analyses were done in one replicate or in another. We never pool biological samples of several animals for our molecular analyses. In the case of the body composition by NMR (**Figures 3c-d**; former Figures 2c-d), the low n (3-5) was due to the osmotic pumps, which interfered with the resonance; therefore, we decided to remove the pumps before the assessment, which was not easy, as it damaged the surrounding fat tissue. Therefore, although we had initially 6-7 animals/group, we only included in the analyses those in which the extraction of the pump did not harm either the WAT or the mice. In any case, following the advice of the Reviewer we have performed some additional experiments increasing the number of animals in the next experimental analyses:

- Body composition of WT and KOR KO mice treated with nicotine (**Figures 3c-d**; n=10-13 WT mice and n=8 KOR KO mice)
- Fat pads of WT and KOR KO mice treated with nicotine (**Figures 3e-f**; n=11-13 WT mice and n=7-9 KOR KO mice)
- ICH UCP1 of WT and KOR KO mice treated with nicotine (**Figures 4e-f**; n=10-12 WT mice and n=11-14 KOR KO mice)
- HE staining of WT and KOR KO mice treated with nicotine (**Figures 4g-h**; n=10-13 WT mice and n=11-14 KOR KO mice)
- Energy expenditure in rats treated with nicotine (**Suppl. Figure 1c**; n=13 vehicle-treated and n=13 nicotine-treated)
- Locomotor activity in rats treated with nicotine (**Suppl. Figure 1d**; n=11 vehicle-treated and n=11 nicotine-treated)
- Respiratory quotient in rats treated with nicotine (**Suppl. Figure 1e**; n=13 vehicle-treated and n=13 nicotine-treated)

We thank the Reviewer for this comment. In this sense, we would like to highlight that as part of this submission, and following *Nature Communications* Editorial Policy, we have uploaded onto the manuscript tracking system 8 Excel files (one per Figure and Suppl. Figure) which contain the source data and original images for any single panel.

The Editor and Reviewers can easily check there the results, statistics, number of animals/subjects etc.

Discussion

Reviewer's comment 6: *The discussion is very limited but so is space, although some of the text could be shortened as it is only summarizing the results. Both in the introduction and the discussion, $\alpha 2$ nAChR is mentioned but since none of the experiments investigated the effects of $\alpha 2$ modulation, the discussion around this receptor is largely irrelevant. Instead, it would be preferable to extend the discussion on UCP1.*

Response: We thank the Reviewer for this comment. In the new version, the discussion has been re-focused on issues related to browning, sympathetic innervation and UCP1 as suggested, removing the parts on $\alpha 2$ -nAChR. However, we have maintained the references about $\alpha 2$ -nAChR in the Introduction for an overall overview of the topic.

We thank again the Reviewer#3 for his/her excellent insight that has improved the scope and quality of our manuscript.

REFERENCES

1. Plum,L. *et al.* Enhanced leptin-stimulated Pi3k activation in the CNS promotes white adipose tissue transdifferentiation. *Cell Metab* **6**, 431-445 (2007).
2. Tews,D. *et al.* FTO deficiency induces UCP-1 expression and mitochondrial uncoupling in adipocytes. *Endocrinology* **154**, 3141-3151 (2013).
3. Ruan,H.B. *et al.* O-GlcNAc Transferase Enables AgRP Neurons to Suppress Browning of White Fat. *Cell* **159**, 306-317 (2014).
4. Neinast,M.D. *et al.* Activation of natriuretic peptides and the sympathetic nervous system following Roux-en-Y gastric bypass is associated with gonadal adipose tissues browning. *Mol. Metab* **4**, 427-436 (2015).
5. Li,G. *et al.* Intermittent Fasting Promotes White Adipose Browning and Decreases Obesity by Shaping the Gut Microbiota. *Cell Metab* **26**, 672-685 (2017).
6. Bauters,D. *et al.* Loss of ADAMTS5 enhances brown adipose tissue mass and promotes browning of white adipose tissue via CREB signaling. *Mol. Metab* **6**, 715-724 (2017).
7. Contreras,C. *et al.* Reduction of Hypothalamic ER Stress Activates Browning of White Fat and Ameliorates Obesity. *Diabetes* **66**, 87-99 (2017).
8. Martinez-Sanchez,N. *et al.* Thyroid hormones induce browning of white fat. *J. Endocrinol.* **232**, 351-362 (2017).
9. Linares-Pose,L. *et al.* Genetic Targeting of GRP78 in the VMH Improves Obesity Independently of Food Intake. *Genes (Basel)* **9**, (2018).
10. Seoane-Collazo,P. *et al.* SF1-Specific AMPK α 1 Deletion Protects Against Diet-Induced Obesity. *Diabetes* **67**, 2213-2226 (2018).
11. López,M. *et al.* Hypothalamic AMPK and fatty acid metabolism mediate thyroid regulation of energy balance. *Nat. Med.* **16**, 1001-1008 (2010).
12. Martínez de Morentin,P.B. *et al.* Estradiol regulates brown adipose tissue thermogenesis via hypothalamic AMPK. *Cell Metab* **20**, 41-53 (2014).
13. Beiroa,D. *et al.* GLP-1 Agonism Stimulates Brown Adipose Tissue Thermogenesis and Browning Through Hypothalamic AMPK. *Diabetes* **63**, 3346-3358 (2014).
14. Martins,L. *et al.* A Functional Link between AMPK and Orexin Mediates the Effect of BMP8B on Energy Balance. *Cell Rep.* **16**, 2231-2242 (2016).
15. Martínez-Sánchez,N. *et al.* Hypothalamic AMPK-ER stress-JNK1 axis mediates the central actions of thyroid hormones on energy balance. *Cell Metab* **26**, 212-229 (2017).
16. Martínez de Morentin,P.B. *et al.* Nicotine induces negative energy balance through hypothalamic AMP-activated protein kinase. *Diabetes* **61**, 807-817 (2012).
17. Seoane-Collazo,P. *et al.* Nicotine improves obesity and hepatic steatosis and ER stress in diet-induced obese male rats. *Endocrinology* **155**, 1679-1689 (2014).
18. Nedergaard,J. & Cannon,B. The browning of white adipose tissue: some burning issues. *Cell Metab* **20**, 396-407 (2014).

19. Morrison,S.F., Madden,C.J., & Tupone,D. Central neural regulation of brown adipose tissue thermogenesis and energy expenditure. *Cell Metab* **19**, 741-756 (2014).
20. Contreras,C. *et al.* The brain and brown fat. *Ann. Med.* **47**, 150-168 (2015).
21. Zeng,W. *et al.* Sympathetic neuro-adipose connections mediate leptin-driven lipolysis. *Cell* **163**, 84-94 (2015).
22. Pirzgalska,R.M. *et al.* Sympathetic neuron-associated macrophages contribute to obesity by importing and metabolizing norepinephrine. *Nat. Med.* **23**, 1309-1318 (2017).
23. Chi,J., Crane,A., Wu,Z., & Cohen,P. Adipo-Clear: A Tissue Clearing Method for Three-Dimensional Imaging of Adipose Tissue. *J. Vis. Exp.*(2018).
24. Chi,J. *et al.* Three-Dimensional Adipose Tissue Imaging Reveals Regional Variation in Beige Fat Biogenesis and PRDM16-Dependent Sympathetic Neurite Density. *Cell Metab* **27**, 226-236 (2018).
25. Munoz-Manchado,A.B. *et al.* Chronic and progressive Parkinson's disease MPTP model in adult and aged mice. *J. Neurochem.* **136**, 373-387 (2016).
26. Franco,R. *et al.* Hints on the Lateralization of Dopamine Binding to D1 Receptors in Rat Striatum. *Mol. Neurobiol.* **53**, 5436-5445 (2016).
27. Garrido-Gil,P., Rodriguez-Perez,A.I., Dominguez-Meijide,A., Guerra,M.J., & Labandeira-Garcia,J.L. Bidirectional Neural Interaction Between Central Dopaminergic and Gut Lesions in Parkinson's Disease Models. *Mol. Neurobiol.* **55**, 7297-7316 (2018).
28. Cruces-Sande,A. *et al.* Copper Increases Brain Oxidative Stress and Enhances the Ability of 6-Hydroxydopamine to Cause Dopaminergic Degeneration in a Rat Model of Parkinson's Disease. *Mol. Neurobiol.* **56**, 2845-2854 (2019).
29. Czyzyk,T.A. *et al.* kappa-Opioid receptors control the metabolic response to a high-energy diet in mice. *FASEB J.* **24**, 1151-1159 (2010).
30. Whittle,A.J. *et al.* Bmp8b increases brown adipose tissue thermogenesis through both central and peripheral actions. *Cell* **149**, 871-885 (2012).

REVIEWERS' COMMENTS:

Reviewer #2 (Remarks to the Author):

All points I raised in the previous round of review have been satisfactorily addressed by the Authors. I think the paper can now be published.

Reviewer #3 (Remarks to the Author):

The authors have responded to my comments and recommendations, and I am satisfied by their answers. The new experiments and data further reinforce this very elegant and comprehensive study. As a side note, I would recommend to add, in the Methods section, the full details of the antibodies used for the nAChRs Western blots as results of experiments with such antibodies can be difficult to interpret (supp Fig 2a).

REVIEWER#2

Overall comment: All points I raised in the previous round of review have been satisfactorily addressed by the Authors. I think the paper can now be published.

Response: We thank the Reviewer for the positive view of our manuscript. We also thank the Reviewer for his/her comments which definitely improved the quality of our study.

REVIEWER#3

Overall comment: The authors have responded to my comments and recommendations, and I am satisfied by their answers. The new experiments and data further reinforce this very elegant and comprehensive study. As a side note, I would recommend to add, in the Methods section, the full details of the antibodies used for the nAChRs Western blots as results of experiments with such antibodies can be difficult to interpret (supp Fig 2a).

Response: We thank the Reviewer for the positive view of our manuscript. We also thank the Reviewer for his/her comments which definitely improved the quality of our study. Following the Reviewer's suggestion, we have added all the information about the antibodies against nAChRs.